# Synthesis and Characterization of Dithiooxamidate-Bridged Polynuclear Ni Complexes

**Tomohiko Hamaguchi ***, **Ryo Kuraoka, Takumi Yamamoto, Naoya Takagi, Isao Ando and Satoshi Kawata**

Department of Chemistry, Faculty of Science, Fukuoka University, Fukuoka 814-0180, Japan; kawata@fukuoka-u.ac.jp (S.K.)
* Correspondence: thama@fukuoka-u.ac.jp

**Abstract:** Mixed-valence complexes contain two metals with different formal oxidation numbers and, therefore, show mixed properties that are influenced by the electronic coupling between the two metals, which is, in turn, regulated by a bridging ligand. This is an attractive point for many researchers. Oxalate is widely used as a bridging ligand for preparing polynuclear complexes. More than 1000 complexes have been reported until now. However, dithiooxamidate, which is an oxalate analog, is less popular as a bridging ligand. Here, a new dithiooxamidate-bridged Ni-diphosphine dinuclear complex with the formula $[(\mu_2\text{-toxa})\{Ni(dppe)\}_2](BF_4)_2$ (toxa = dithiooxamidate; dppe = 1,2-bis(diphenylphosphino)ethane) was prepared and characterized via single-crystal X-ray diffraction. When using 1,3-bis(diphenylphosphino)propane (dppp) instead of dppe, dinuclear, trinuclear, and tetranuclear complexes were obtained, i.e., $[(\mu_2\text{-toxa})\{Ni(dppp)\}_2](BF_4)_2$, $[\{\mu_2\text{-Ni(toxa)}_2\}\{Ni(dppp)\}_2](BF_4)_2$, and $[\{\mu_3\text{-Ni(toxa)}_3\}\{Ni(dppp)\}_3](BF_4)_2$, respectively. Bidentate toxa ligands in dinuclear complexes coordinate each Ni atom as $\kappa(S,N)$. However, the trinuclear and the tetranuclear complexes have the toxa ligands with $\kappa(N,N)$ and $\kappa(S,S)$ coordination. The $[(\mu_2\text{-toxa})\{Ni(dppe)\}_2](BF_4)_2$ complex undergoes four reversible redox processes, whose analysis via a controlled-potential absorption spectrum reveals the presence of a Ni(II)-Ni(I) mixed-valence state at $\Delta E_{1/2} = 0.22$ V with a comproportionation constant of $6.1 \times 10^3$.

**Keywords:** Ni complex; dithiooxamide; polynuclear complex; electrochemistry

## 1. Introduction

Mixed-valence complexes have attracted considerable interest since long, leading to numerous reports [1–5]. These complexes contain two metals with different formal oxidation numbers and, therefore, show mixed properties that are influenced by the electronic coupling between the two metals, which is, in turn, regulated by a bridging ligand. In addition, the regulation of the electronic coupling is also achieved by external stimulation (e.g., temperature, pressure, and light), which results in valence tautomerism [6,7].

Oxalate is widely used as a bridging ligand for preparing polynuclear complexes. In contrast, oxamidate and dithiooxamidate, which are oxalate analogs, are less popular as bridging ligands. For example, a search in the CCDC database (updated in November 2022) [8] results in more than 1000 polynuclear complexes bearing oxalate as the bridging ligand, whereas the number of hits for polynuclear complexes with bridging oxamidate and substituted oxamidate ligands, dithiooxamidate and substituted dithiooxamidate ligands is 500 and 30, respectively. Only 12 polynuclear complexes with bridging unsubstituted dithiooxamidate ligands are reported [9–17].

To investigate the relationship between the electronic coupling between metal centers and the bridging ligand, several groups studied dinuclear complexes with oxalate analogs. For example, Cotton et al. reported the electronic coupling between $Mo_2$ dimetal units in oxamidate-bridged and dithiooxamidate-bridged bis-$(Mo_2)$ complexes [14]. Mikhalyova et al. investigated the exchange coupling in oxalate-bridged, oxamidate-bridged, and

dithiooxamidate-bridged dinuclear complexes [15]. Similarly, we aimed to prepare a series of dinuclear Ni complexes bearing oxalate, oxamidate, and dithiooxamidate, respectively. Unfortunately, we only obtained dithiooxamidate–Ni dinuclear complexes.

Here, we report the synthesis and X-ray single-crystal structure of a dithiooxamidate-bridged Ni polynuclear complexes, i.e., [($\mu_2$-toxa){Ni(dppe)}$_2$](BF$_4$)$_2$, [($\mu_2$-toxa){Ni(dppp)}$_2$] (BF$_4$)$_2$, [{$\mu_2$-Ni(toxa)$_2$}{Ni(dppp)}$_2$](BF$_4$)$_2$, and [{$\mu_3$-Ni(toxa)$_3$}{Ni(dppp)}$_3$](BF$_4$)$_2$ (H$_2$toxa = dithiooxamide; dppe = 1,2-bis(diphenylphosphino)ethane; dppp = 1,3-bis (diphenylphosphino) propane. Figure 1). The dithiooxamidate shows different coordination between dinuclear complexes and others. We also report electrochemical study of [($\mu_2$-toxa){Ni(dppe)}$_2$](BF$_4$)$_2$.

**Figure 1.** Molecular structures of ligands.

## 2. Materials and Methods

All materials were purchased from commercial suppliers (FUJIFILM Wako Pure Chemical Corporation (Osaka, Japan); Kanto Chemical Co., Inc. (Tokyo, Japan); and Tokyo Chemical Industry Co., Ltd. (Tokyo, Japan)) and used as received without further purification.

### 2.1. Measurements

Absorption spectra were measured using a SHIMADZU UV-3600 UV-vis-NIR spectrophotometer (Kyoto, Tapan). $^1$H NMR (400 MHz) spectra were measured in CD$_3$CN with a Bruker Avance III HD spectrometer (Rheinstetten, Germany) using CD$_3$CN ($\delta$ = 1.94 ppm) as an internal standard. $^{31}$P NMR (161 MHz) spectra were measured in CD$_3$CN with a Bruker Avance III HD spectrometer (Rheinstetten, Germany) using 85% H$_3$PO$_4$ in water ($\delta$ = 0.00 ppm) as an external standard. Cyclic voltammetry was performed on a BAS BAS100B/W electrochemical workstation (West Lafayette, IN, USA) using a three-electrode cell with a glassy carbon electrode as the working electrode, a Pt coil electrode as the counter electrode, and a homemade Ag$^+$/Ag electrode as the reference electrode. The

$E_{1/2}$ of the ferrocenium/ferrocene couple was 0.11 V vs. $Ag^+/Ag$. The medium was 0.1 mol $dm^{-3}$ $n$-$Bu_4NPF_6$/$CH_3CN$, and the concentration of the complex was $1 \times 10^{-3}$ mol $dm^{-3}$. Controlled-potential absorption spectra were obtained using an optically transparent thin-layer electrode cell and a HOKUTO DENKO HABF501 Potentiostat Galvanostat (Tokyo, Japan). The working electrode was a Pt mesh (80 mesh); the counter electrode was a Pt coil, and the reference electrode was a homemade $Ag^+/Ag$ electrode. The optical path length was 1 mm. The medium was 0.1 mol $dm^{-3}$ $n$-$Bu_4NPF_6$/$CH_3CN$. The concentration of the complex was $3.91 \times 10^{-4}$ mol $dm^{-3}$. All electrochemical measurements were performed at room temperature (20 °C) under a nitrogen atmosphere. C, H, and N analyses were conducted by the Service Center for the elementary analysis of organic compounds of Kyushu University.

## 2.2. Synthesis Procedures

### 2.2.1. Synthesis of [($\mu_2$-toxa){Ni(dppe)}$_2$](BF$_4$)$_2$ (**1**)

Dppe (586 mg, 1.47 mmol) was added to a solution of [Ni(H$_2$O)$_6$](BF$_4$)$_2$ (500 mg, 1.47 mmol) in $CH_3CN$ (20 mL), and the mixture was stirred for 3 h. To the resulting yellow suspension, dithiooxamide (H$_2$toxa; 88.4 mg, 0.734 mmol) and Et$_3$N (200 μL, 1.47 mmol) were added, and the mixture was stirred overnight. The resulting dark brown solution was evacuated to dryness; the residual solid was dissolved with a small amount of $CH_3CN$, and an excess amount of diethyl ether was added to the solution. The resulting precipitate was collected by filtration and dried in vacuo, affording complex **1** in a 72.5% yield (642 mg). Elemental analysis (%) found: C, 52.45; H, 4.19; N, 2.86. Calc. for $C_{54}H_{50}B_2F_8N_2Ni_2P_4S_2 \cdot (H_2O)_2 \cdot (CH_3CN)_{0.5}$: C, 52.32; H, 4.43; N, 2.77. Absorption data ($CH_3CN$) $\lambda$/nm($\varepsilon$/dm$^3$ mol$^{-1}$ cm$^{-1}$): 450sh (4800); 380sh (9500); 310sh (29600); 273 (43200) $^1$H NMR $\delta$ 8.23 (s, 2H, N–H); 7.77 (br, 16H, $o$-phenyl); 7.68 (t, 8H, $J$ = 7.6 Hz, $p$-phenyl); 7.56 (t, 16H, $J$ = 7.6 Hz, $m$-phenyl); 2.50 (m, 8H, PC$H_2$C$H_2$P). $^{31}$P{$^1$H} NMR $\delta$ 62.64 (br). The $^1$H NMR and $^{31}$P NMR spectra are shown in Figures S1 and S2, respectively.

### 2.2.2. Synthesis of [($\mu_2$-toxa){Ni(dppp)}$_2$](BF$_4$)$_2$ (**1a**), [{$\mu_2$-Ni(toxa)$_2$}{Ni(dppp)}$_2$](BF$_4$)$_2$ (**2a**), [{$\mu_3$-Ni(toxa)$_3$}{Ni(dppp)}$_3$](BF$_4$)$_2$ (**3a**)

To a solution of [Ni(H$_2$O)$_6$](BF$_4$)$_2$ (200 mg, 0.588 mmol) in $CH_3CN$ (10 mL) was added dppp (242 mg, 0.588 mmol), and the mixture was stirred for 3 h. H$_2$toxa (35.0 mg, 0.294 mmol) and Et$_3$N (41 μL, 0.59 mmol) were added to the resulting brown solution, and the mixture was stirred overnight. Subsequent evacuation to dryness and recrystallization of the red-brown crude product via vapor diffusion of diethyl ether into a $CH_3CN$ solution afforded brown block crystals along with dark-blue muddy precipitates. The brown block crystals were a mixture of complexes **1a**, **2a**, and **3a**.

## 2.3. X-ray Crystallography

Single crystals of **1** suitable for single-crystal X-ray analysis were obtained by slow recrystallization via vapor diffusion of diethyl ether into a $CH_3CN$ solution of the complex at room temperature. Single crystals of **1a** and **2a** for X-ray analysis were obtained, following the synthesis method described in Section 2.2.2. Single crystals of **3a** for X-ray analysis were obtained via vapor diffusion of diethyl ether into an acetone solution of a mixture of **1a**, **2a**, and **3a** at room temperature. The data were collected on a Rigaku R-AXIS RAPID II diffractometer (Tokyo, Japan) for complexes **1**, **1a**, and **2a** and a Rigaku Saturn 724 CCD area-detector diffractometer for complex **3a**. An absorption correction was applied to the intensity data. The structure was solved using a direct method (*SHELXT-2014/5*) [18] and refined by the full-matrix least-squares method on $F^2$ (*SHELXL-2016/6*) [18] using the Yadokari-*XG* software package [19]. All nonhydrogen atoms were refined with anisotropic parameters. H atoms of toxa$^{2-}$ except for complex **1a** and water were located in a difference Fourier map, and the coordinate was fixed. Other H atoms were included in the calculated positions and refined using a riding model. Crystallographic diagrams were created using the *ORTEP* program [20]. Complex **1a** seemed to have one diethyl ether and two $CH_3CN$ molecules as

solvent molecules; however, they were badly disordered and showed large temperature factors. Therefore, they were removed from the final refinement performed using the *SQUEEZE* program in the *PLATON* program [21]. The toxa ligands of complex **1a** are disordered over two positions (N1a/S1a%/C28/C28%/S1a/N1a% and S2b/N2b%/C28/C28%/N2b/S2b%, N3a/S3a$/C56/C56$/S3a/N3a$ and S4b/N4b$/C56/C56$/N4b/S4b$) and the occupancies are 0.422:0.578 (N1a/S1a%/C28/C28%/S1a/N1a%:S2b/N2b%/C28/C28%/N2b/S2b%) and 0.356:0.644 (N3a/S3a$/C56/C56$/S3a/N3a$:S4b/N4b$/C56/C56$/N4b/S4b$), as a result of optimization. One acetonitrile (N7B-C90B-C91B) and triethylamine in complex **2a** are treated as a disorder, and the occupancy is 0.570:0.430 (acetonitrile:triethylamine) as a result of optimization. A phenyl group of a dppp in complex **3a** is disordered over two positions (C70A-C71A-C72-C73A-C74A-C75A and C73B-C74B-C75B-C72-C76B-C77B) and the occupancy is 0.604:0.396 (C70A-C71A-C72-C73A-C74A-C75A:C73B-C74B-C75B -C72-C76B-C77B), as a result of optimization. One acetone in complex **3a** is also disordered over two positions (O6A-C105-C106-C107 and O6B-C108-C106-C107), and the occupancy is 0.531:0.469 (O6A-C105-C106-C107:O6B-C108-C106-C107), as a result of optimization. Crystallographic data are summarized in Table S1. The CCDC 2259972, 2259973, 2259974, and 2259975 contain the crystallographic data **1**, **1a**, **2a**, and **3a** of this paper.

## 3. Results and Discussion

### 3.1. Synthesis of μ-Toxa Polynuclear Ni Complexes

The $\mu_2$-toxa dinuclear Ni complex [($\mu_2$-toxa){Ni(dppe)}$_2$](BF$_4$)$_2$ (**1**) was prepared via a one-pot reaction using [Ni(H$_2$O)$_6$](BF$_4$)$_2$, dppe, H$_2$toxa, and Et$_3$N. The reaction afforded a brown crude product, from which the complex was obtained via recrystallization with CH$_3$CN/diethyl ether.

In the case of using dppp instead of dppe, a similar one-pot reaction and evacuation produced similar red-brown crude. However, a treatment of a small amount of CH$_3$CN and the excess amount of diethyl ether produced red-brown sticky solids. To purify the sticky solids, slow vapor recrystallization was carried out. The recrystallization afforded brown block crystals along with dark-blue muddy precipitates. The brown block crystals were mixture of the dinuclear [($\mu_2$-toxa){Ni(dppp)}$_2$](BF$_4$)$_2$ (**1a**), trinuclear [{$\mu_2$-Ni(toxa)$_2$}{Ni(dppp)}$_2$](BF$_4$)$_2$ (**2a**), and tetranuclear [{$\mu_3$-Ni(toxa)$_3$}{Ni(dppp)}$_3$](BF$_4$)$_2$ (**3a**). In the reaction, the main product was the muddy precipitates, and the yield of the complexes was very low. We speculate that prolonged recrystallization with polar solvent might break complex **1a** and might restructure **1a** to produce dark-blue muddy precipitates in **2a** and **3a**. An appropriate synthesis method affording good yields of **1a**, **2a**, and **3a** has not been established yet.

### 3.2. Structure of μ-Toxa Polynuclear Ni Complexes

The crystal structure of complex **1** is shown in Figure 2. The complex consists of one complex cation, two BF$_4^-$ counter anions, two CH$_3$CN, two water molecules, and one dppe dioxide. The cation has two mono-Ni(dppe) units bridged by a toxa ligand, forming the dinuclear Ni complex. The possible bridging modes of the toxa ligand between two metals are shown in Figure 3. The toxa ligand in complex **1** is coordinated in a trans form. Several μ-toxa and μ-substituted-toxa dinuclear structures have been reported [10,13,16,22–24]. The difference Fourier map of complex **1** revealed the presence of one proton in each N atom; therefore, deprotonation of dithiooxamide resulted in the loss of one proton from each N atom. Each Ni atom exhibits a P/P/N/S four-coordination environment constructed by the coordination of a bidentate dppe ligand and a toxa-$\kappa$(*S*,*N*) bridging ligand. The deviation of Ni1 from the least-square mean plane of P1/P2/S1/N1# is 0.03 Å. The dihedral angle between the least-square mean plane of P1/P2/Ni1 and the least-square mean plane of N1#/S1/Ni1 is 9.7(2)°. These data indicate that the Ni(II) ion has a slightly distorted square-planar geometry and two of these structures are arranged in an almost coplanar fashion in complex **1**.

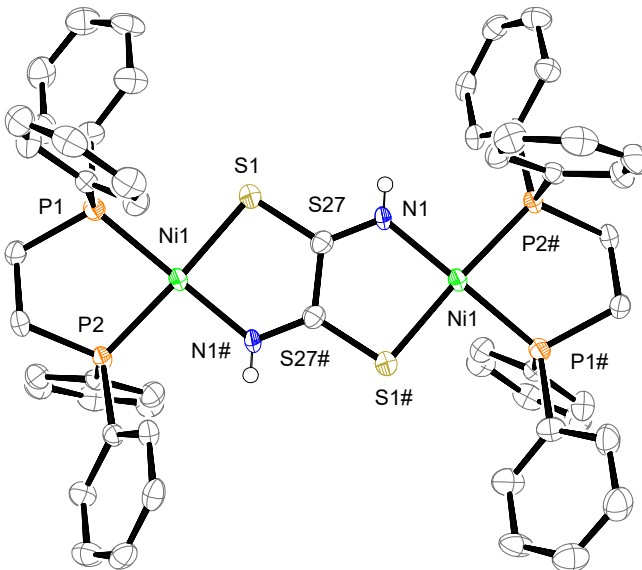

**Figure 2.** Crystal structure of the cation of complex **1**. The counter anions, solvent molecules, diphosphine dioxide, and hydrogen atoms, except for N–H, are omitted for clarity. The thermal ellipsoids are drawn at the 50% probability level. The subscripts # indicate the equivalent atoms generated by the symmetry operators (1-x, 1-y, -z).

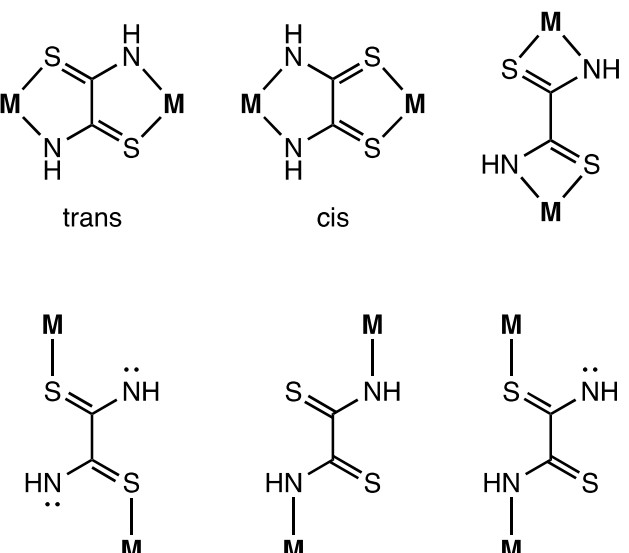

**Figure 3.** Bridging modes of the dithiooxamidate ligand.

The crystal structure of complex **1a** is shown in Figure 4, which displays two crystallographically independent molecules. Complex **1a** consists of one complex cation, two $BF_4^-$ counter anions, and some solvent molecules; however, the solvent molecules were removed from the final refinement performed using the *SQUEEZE* program. The toxa ligands are disordered over two positions. Protons were not detected over N atoms in the difference Fourier map; therefore, they were treated using a riding model. The cation showed almost the same structure as μ-toxa dinuclear Ni-dppe complex **1**. The deviation of Ni from the least-square mean plane of P/P/S/N is 0.08, 0.06, 0.37, and 0.38 Å. The dihedral angle between the least-square mean plane of P/P/Ni and the least-square mean plane of N/S/Ni is 6.5(6)°, 6.2(5)°, 26.0(3)°, and 22.7(2)°. In general, the Ni(II) ion has a slightly distorted square-planar geometry. The dissimilarities between these values can be attributed to the disorder of the toxa ligand. The two four-coordinated square-planar Ni structures are almost coplanar.

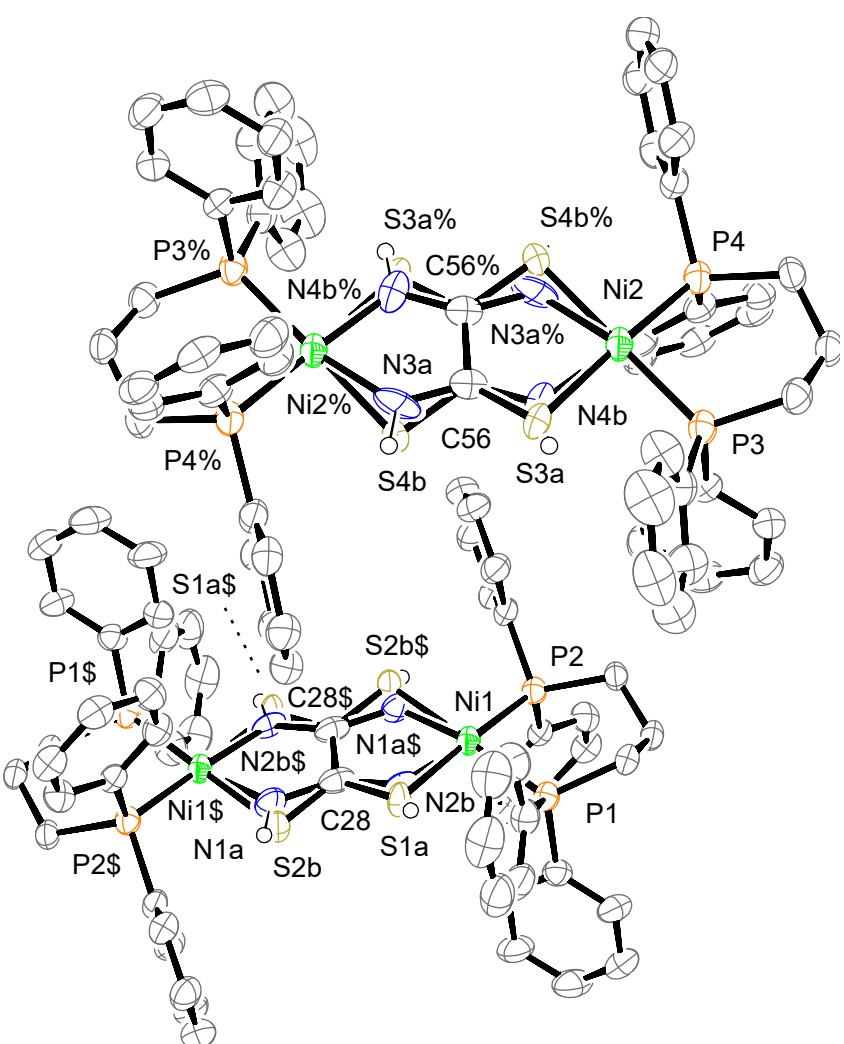

**Figure 4.** Crystal structure of the cation of complex **1a**. The counter anions and hydrogen atoms, except for N–H, are omitted for clarity. The thermal ellipsoids are drawn at the 50% probability level. The subscripts $ and % indicate the equivalent atoms generated by the symmetry operators (2-x, 2-y, 1-z) and (1-x, 1-y, 2-z), respectively.

The crystal structure of complex **2a** is shown in Figure 5. The complex consists of one complex cation, two $BF_4^-$ counter anions, one dppp dioxide, and some molecules (2.43 $CH_3CN \cdot 0.57$ $Et_3N$). The cation contains two mono-Ni(dppp) units bridged by a [Ni{toxa-$\kappa(N,N)$}$_2$]$^{2-}$ moiety instead of a toxa ligand, forming a trinuclear Ni complex. The toxa ligands bridge two Ni atoms in a cis form, which differs from the bridging mode of the toxa ligand in complex **1a**. Although similar μ-toxa and μ-substituted-toxa trinuclear structures have been reported [12,25–27], our complex shows a unique structure for a homonuclear Ni complex. Two terminal Ni atoms (Ni1, Ni2) are coordinated by the S atoms of toxa and the P atoms of dppp. The deviation of the terminal Ni atoms from the least-square mean plane of P/P/S/S is 0.08 and 0.06 Å. The deviation of the central Ni atom (Ni3) from the least-square mean plane of N/N/N/N is 0.09 Å. The dihedral angles between the least-square mean plane of P/P/terminal Ni and the least-square mean plane of S/S/terminal Ni are 6.2° and 4.3(1)°. The dihedral angle between the least-square mean plane of N1/N2/central Ni and the least-square mean plane of N3/N4/central Ni is 9.1(1)°. As observed for complexes **1** and **1a**, the Ni(II) ion exhibits a slightly distorted square-planar geometry.

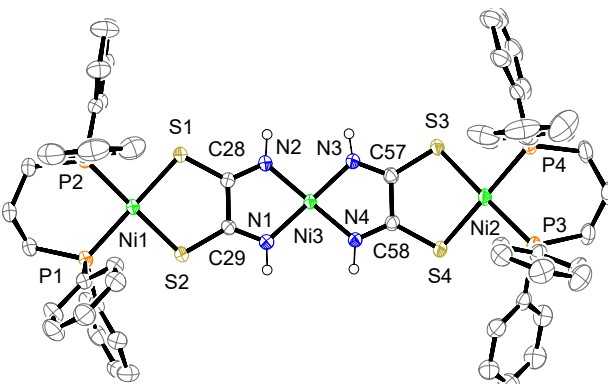

**Figure 5.** Crystal structure of the cation of complex **2a**. The counter anions, solvent molecules, diphosphine dioxide, triethylamine, and hydrogen atoms, except for N–H, are omitted for clarity. The thermal ellipsoids are drawn at the 50% probability level.

The crystal structure of complex **3a** is shown in Figure 6. The complex consists of one complex cation, two $BF_4^-$ counter anions, and six acetone molecules. One phenyl group bonded to P6 is disordered over two positions. The cation has three mono-Ni (dppp) units bridged by a $[Ni\{toxa-\kappa(N,N)\}_3]^{4-}$ moiety, which results in a tetranuclear Ni complex. As in the case of complex **2a**, the toxa ligands bridge two Ni atoms in a cis form. The central Ni atom (Ni4) is coordinated by six N atoms of toxa ligands, resulting in an octahedral geometry. Three terminal Ni atoms (Ni1, Ni2, and Ni3) are coordinated by the S atoms of toxa and the P atoms of dppp. The deviation of the terminal Ni atoms from the least-square mean plane of P/P/S/S is 0.05, 0.07, and 0.04 Å. The dihedral angles between the least-square mean plane of P/P/terminal Ni and the least-square mean plane of S/S/terminal Ni are 12.9(1)°, 5.1(1)°, and 9.0(1)°. As observed for the other complexes, the terminal Ni(II) ion shows a slightly distorted square-planar geometry.

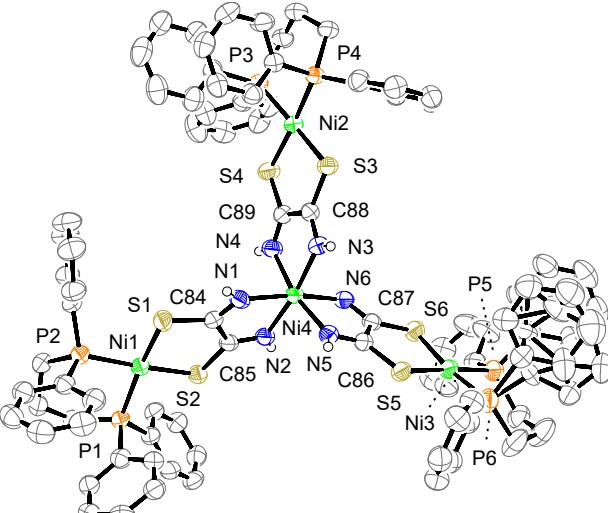

**Figure 6.** Crystal structure of the cation of complex **3a**. The counter anions, solvent molecules, and hydrogen atoms, except for N–H, are omitted for clarity. The thermal ellipsoids are drawn at the 50% probability level.

Selected bond distances and bond angles are summarized in Table S2. The four complexes show similar Ni–S and Ni–P bond distances but different Ni–N bond distances. Thus, the Ni–N bond distance of Ni-dppp complex **1a** is shorter than that of Ni-dppe complex **1**, which could be due to the different diphosphine ligands. Comparison of Ni–N distance with complex **1a** and complex **2a** makes us understand the influence of

the coordination mode (trans vs. cis). In this case, there is no significant difference. Meanwhile, complex **3a** shows the longest Ni–N distance, which could be due to the different coordination modes (octahedral vs. square-planar). The P–Ni–P bond angle reflects the difference between dppe and dppp. The N–Ni–S angle is almost the same in complexes **1** and **1a**. The S–Ni–S angle is almost the same in complexes **2a** and **3a**; however, the N–Ni–N angle of complex **3a** is narrower than that of complex **2a**. This difference could be due to the different coordination modes, as observed for the Ni–N bond distance. Bridging toxa ligand is also surveyed by bond distance. The CCDC database shows that the N–C, S–C, and C–C bond lengths of μ-toxa in polynuclear complexes are 1.22 [17]–1.48 [12], 1.33 [17]−1.80 [14], and 1.43 [16]−1.58 [9] Å, respectively, with respective median values of 1.29 [14,28], 1.72 [29,30], and 1.50 [15,30] Å. Therefore, the bond lengths of the toxa ligand in the complexes prepared in this study fall within the range of those previously reported for polynuclear complexes.

### 3.3. Effect of Ancillary Diphosphine Ligand for the Product

The synthesis of the toxa-bridged Ni complexes involves two steps. $[Ni(H_2O)_6](BF_4)_2$ and diphosphine are mixed in $CH_3CN$ in the first step, which produces $[Ni(diphosphine)(CH_3CN)_2]^{2+}$. In the second step, $H_2toxa$ and $Et_3N$ are added to the solution, and $toxa^{2-}$ ligand coordinates Ni atoms of $[Ni(diphosphine)(CH_3CN)_2]^{2+}$. The bite angle of the diphosphine ligand is known as an important effect for a property of complexes [31]. The difference between dppe and dppp could invoke the different affinity between the Ni atom and the S/N atoms of toxa. We think that the affinity might produce the difference in the products; however, we could not prove it experimentally in this work.

### 3.4. Electrochemical Behavior of Dinuclear Ni Complex **1**

Cyclic voltammetry was conducted to estimate the electronic coupling between Ni atoms through the toxa ligand in complex **1**. As shown in Figure 7, the cyclic voltammogram recorded at a scan rate of 100 mV s$^{-1}$ displays four quasi-reversible one-electron redox couples and two oxidation peaks, as summarized in Table 1. At higher scan rates, the quasi-reversible redox couples turned more reversible, and the oxidation peaks became more obscure. Overall, complex **1** shows four reversible one-electron redox couples.

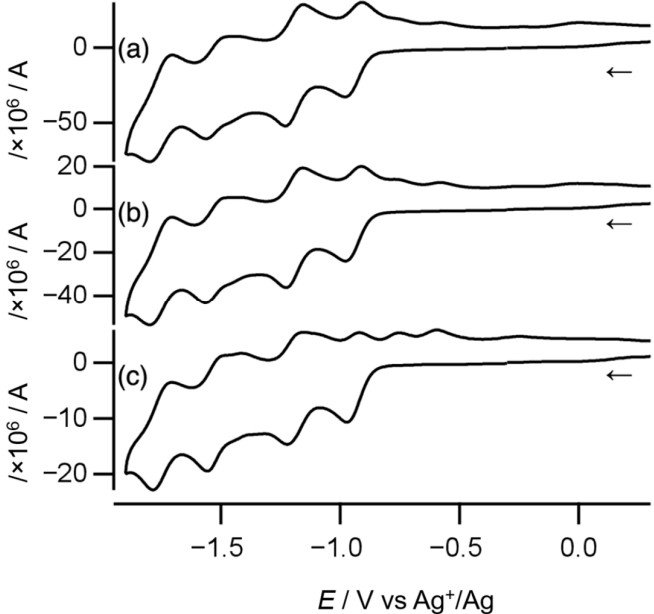

**Figure 7.** Cyclic voltammograms of complex **1** recorded at scan rates of (**a**) 1000 mV s$^{-1}$, (**b**) 500 mV s$^{-1}$, and (**c**) 100 mV s$^{-1}$. The arrows indicate the direction of the initial scan.

**Table 1.** Electrochemical behavior of complex **1** under various scan rates.

| | | | | Scan Rate/mV s$^{-1}$ | | | | |
| | **100** | | | **500** | | | **1000** | |
| $E_{pa}$/V | $E_{pc}$/V | $E_{1/2}$/V | $E_{pa}$/V | $E_{pc}$/V | $E_{1/2}$/V | $E_{pa}$/V | $E_{pc}$/V | $E_{1/2}$/V |
|---|---|---|---|---|---|---|---|---|
| −1.71 | −1.79 | −1.75 | −1.71 | −1.80 | −1.76 | −1.70 | −1.80 | −1.75 |
| −1.50 | −1.56 | −1.53 | −1.50 | −1.57 | −1.54 | −1.48 | −1.56 | −1.52 |
| −1.41 | | | −1.41 | | | | | |
| −1.16 | −1.22 | −1.19 | −1.16 | −1.23 | −1.20 | −1.16 | −1.23 | −1.20 |
| −0.92 | −0.98 | −0.95 | −0.91 | −0.98 | −0.95 | −0.91 | −0.98 | −0.95 |
| −0.76 | | | −0.75 | | | | | |
| −0.60 | | | −0.58 | | | −0.56 | | |

All redox potentials are reported vs. Ag$^+$/Ag.

Complex **1** contains two Ni atoms and one toxa ligand, which could act as a redox-active center. To identify the four redox couples where the reduction occurs, a controlled-potential absorption spectrum at each reduced state was measured. As shown in Figure 8, only the three-electron reduced state has a peak (8.9 × 10$^3$ cm$^{-1}$, 2.5 × 10$^3$ dm$^3$ mol$^{-1}$ cm$^{-1}$) in the near IR region. Furthermore, the peak disappeared in a four-electron reduced state. The peak would be an intervalence charge transfer band; therefore, the third and fourth reduction processes are due to Ni(II)–Ni(II)/Ni(II)–Ni(I) and Ni(II)–Ni(I)/Ni(I)–Ni(I), respectively, and the first and second reduction processes stem from toxa$^{2-}$/toxa$^{3-}$ and toxa$^{3-}$/toxa$^{4-}$, respectively. As far as we know, this is the first report of a mixed valence state of Ni(II)–Ni(I) bridged by toxa$^{3-}$.

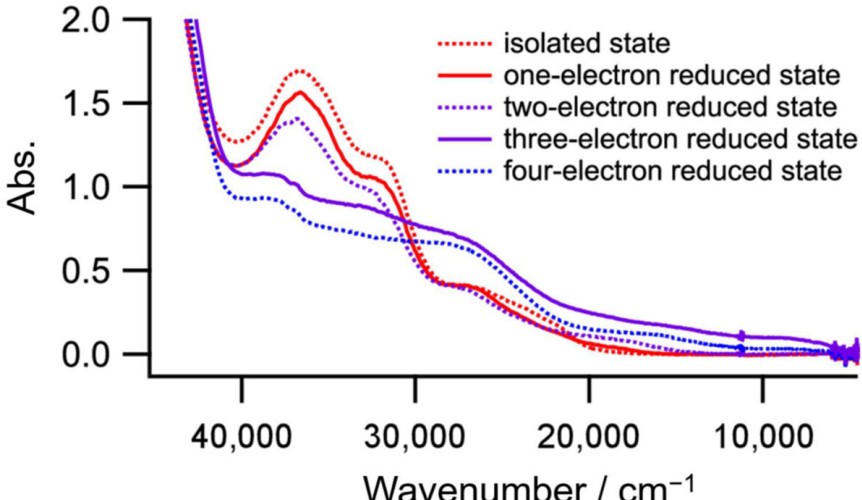

**Figure 8.** Absorption spectra of each reduced state of complex **1**.

The mixed-valence state Ni(II)–Ni(I) occurs at $\Delta E_{1/2}$ = 0.22 V ($E_{1/2}$(Ni(II)–Ni(II)/Ni(II)–Ni(I)) = −1.53 V, $E_{1/2}$(Ni(II)–Ni(I)/Ni(I)–Ni(I)) = −1.75 V). As shown in Equation (1), the separation between the two redox potentials is associated with the comproportionation constant $K_c$ (Equation (1)), which can be calculated using Equation (2) [32]. For complex **1**, $K_c$ was determined to be 6.1 × 10$^3$.

$$[\text{Ni(II)} - \text{Ni(II)}]^{n+} + [\text{Ni(I)} - \text{Ni(I)}]^{(n-2)+} \overset{K_c}{\rightleftharpoons} 2[\text{Ni(II)} - \text{Ni(I)}]^{(n-1)+} \tag{1}$$

$$K_c = exp\left[\frac{(E_1^0 - E_2^0)n_1 n_2 F}{RT}\right] \tag{2}$$

where $E^0{}_1$ and $E^0{}_2$ are the standard redox potentials of the first redox reaction and the second redox reaction, respectively; $n_1$ and $n_2$ are the numbers of electrons at the first redox reaction and the second redox reaction, respectively; $F$ is the Faraday constant; $R$ is the gas constant, and $T$ is the temperature.

## 4. Conclusions

In this study, new toxa-bridged Ni-diphosphine polynuclear complexes were prepared and characterized via X-ray diffraction. Ni-dppe dinuclear complex **1** was obtained from a one-pot reaction with dppe as a diphosphine ligand. When using dppp instead of dppe, dinuclear complex **1a**, trinuclear complex **2a**, and tetranuclear complex **3a** were obtained. The dithiooxamidate shows different coordination between the dinuclear and the trinuclear/tetranuclear complexes. The electrochemical behavior of Ni-dppe dinuclear complex **1** was investigated via cyclic voltammetry and absorption spectra at various potentials, and the electronic coupling between two Ni atoms was discussed.

Further investigation of oxamidate/dithiooxamidate-bridged Ni-diphosphine dinuclear complexes is currently underway in our laboratory, including the optimization of the synthesis method to obtain trinuclear and tetranuclear complexes in good yield to examine the relationship between their molecular structure and their electronic coupling. The synthesis reaction is also explored for the optimization.

**Supplementary Materials:** The following supporting information can be downloaded at https://www.mdpi.com/article/10.3390/chemistry5040150/s1, Figure S1: $^1$H NMR spectrum of complex **1** in CD$_3$CN; Figure S2: $^{31}$P NMR spectrum of complex **2** in CD$_3$CN (85% H$_3$PO$_4$ aq. shows a singlet peak at $\delta -1.458$ ppm); Table S1: Crystallographic data; Table S2: Selected bond lengths (Å) and angles (°).

**Author Contributions:** Conceptualization, T.H.; methodology, T.H.; investigation, T.H., R.K., T.Y. and N.T.; resources, I.A. and S.K.; writing—original draft preparation, T.H.; writing—review and editing, I.A. and S.K.; supervision, I.A. and S.K. All authors have read and agreed to the published version of the manuscript.

**Funding:** This research received no external funding.

**Data Availability Statement:** CCDC 2259972-2259975 contains the supplementary crystallographic data for this paper. These data can be obtained free of charge from The Cambridge Crystallographic Data Centre via www.ccdc.cam.ac.uk/data_request/cif accessed on 7 September 2023. Supplementary Materials associated with this article can be found, in the online version.

**Conflicts of Interest:** The authors declare no conflict of interest.

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
