# Peer review of "Synthesis and Characterization of Dithiooxamidate-Bridged Polynuclear Ni Complexes"

_chemistry, doi:10.3390/chemistry5040150_

Round 1

Reviewer 1 Report

In this work, several polynuclear Ni complexes were synthesized and characterized. The discussion in explaining the difference in coordination mode through DFT calculations is not very convinding in the current version. The computational method is also nor reasonable. This work can be improved and more appealing if the authors consider the following issues.

(1)    SDD basis set was used for Ni in geometry optimization. But 6-31g* was applied for Ni in single point energy calculation. This is not appropriate as 6-31g* contain smaller basis functions than SDD. A larger basis sets should be used for single point calculations to gain accurate energies, even for other light atoms.

(2)    In Figure 1, the atoms should be C27 & C27# instead of S27 & S27#.

(3)    The model structures for possible precursor complexes are not reasonable. The coordination number of N should be 3. But the coordination number of some N atoms for the bottom two structures in Figure 6 is 2, which leading to their instability. But in the synthesized systems, all N/S atoms are coordinated with Ni and there is no unsaturated N atom. Thus, it is not very convincing to use the relative energies of these three possible precursors to reveal the favoring of toxa-(κN,N) in section 3.3.

(4)    In line 270 of page 8, it shows “In this process, μ2-{toxa-(κS,N)} would be the preferred coordination mode owing to the existence of structural symmetry.” Could the author explain how the structural symmetry affect the coordination mode in detail?

(5)    As DFT calculations were performed., it will be very helpful to calculate the atomic charges of atoms in complexes 1, 1a, 2a, and 3a, which can provide more insight in understanding the oxidation states of Ni atoms as well as the structural properties such as the difference in Ni-N bond distance. Currently, the use of DFT calculations in only calculating the possible precursors does not show very useful messages.

Reviewer 2 Report

The paper of Tomohiko Hamaguchi and co-authors is an interesting work on synthesis new Ni(II) dithiooxamidate-bridged complexes. For four new compounds were grown single crystals amenable for structure determination. The structures obtained by the authors were deposited in the Cambridge Structural Database. This is a classic "crystal-structure" type article. For the most part, this article represents the publication of so-called “negative results”, when the authors were unable to develop a method for the synthesis of a phase-pure product. However, the article contains 4 structures deposited in the database, which may lead to additional citations to the Chemistry journal.

The article is well written, I have no major comments, but there are several clarifications and recommendations that, in my opinion, can improve the perception of the article.

Line 28: “Mixed-valence complexes have attracted considerable interest since long, leading to numerous reports [1-5].” - references 1-5 refer to the years 2000-2006, at the same time, I would advise the authors to pay attention to citing more recent literature. For example, it would be worth paying attention to reviews/articles on molecular dynamic systems, which usually contain metal cations in different oxidation states and the phenomenon of redox isomerism (or valence tautomerism) occurs. (see for example 10.1038/nchem.2547, 10.1016/j.ccr.2014.01.014, 10.1021/acs.inorgchem.7b01344)

Line 38: “Only 12 polynuclear complexes with bridging unsubstituted dithiooxamidate ligands are reported [7-15]” – References 7-15 refer to 1983/1984/1993/1995/2007/2022 - the insufficient presence of objects in articles may be the reason either for the difficulty of obtaining and describing them, or for their uselessness (it may sound rude). There are 12 complexes in total - did the authors search the Cambridge Structural Database? Or did authors use the Reaxys and SciFinder databases? Perhaps more than 12 of such complexes are known and they were characterized by other methods of analysis (MALDI-TOF for example?)

Structural formulas of ligands (dppe and dppp) should be added

Line 101 and 109-110: “The brown block crystals were a mixture of complexes 1a, 2a, and 3a” and “Single crystals of 3a for X-ray analysis were obtained via vapor diffusion of diethyl ether into an acetone solution of a mixture of 1a, 2a, and 3a at room temperature.” In my opinion, there is some discrepancy here: after all, 1a, 2a, 3a were obtained using the one-pot method or was product 3a obtained by recrystallization a mixture of them from acetone? Did the authors perform powder diffraction on the sample obtained by recrystallization from acetone? Perhaps the authors were able to isolate pure phase 3a?

How do the authors get the confidence that there are only three products in the reaction with dppp? Did the authors perform powder diffraction and compare the powder patterns with the results of single crystal diffraction? Maybe there are more products? And also, using powder diffraction, it would be possible to evaluate the contribution of each phase to the formation of a powder pattern, thereby estimating the yield of each reaction product.

How are the authors confident that product 1 is the only one in the reaction with dppe? Have authors done powder diffraction and compared the results with the results of single crystal diffraction? NMR is not the best help here, in my humble opinion, since there will likely be one product in the solution.

Table 1, Line 282: in my opinion, the energies of formation are very close, I would not recommend using the phrases “the most stable structure” or “the most unstable structure”. And reference 31 should be removed - this is not a reference, it is confusing. It can be made a footnote for example. In my opinion, the authors should highlight that the reactions of product formation are very close, and that is why “An appropriate synthesis method affording good yields of 1a, 2a, and 3a has not been established yet” (Line 165-166).

I found the part on the electrochemical study of complex 1 interesting. However, the formation of monovalent nickel, in my opinion, has not been fully proven. It would be interesting to observe a simultaneous study using the EPR method, such as in this paper 10.3390/molecules26102998 , to detect the formation of paramagnetic nickel(I).

As I said at the beginning of the review, the manuscript is well written, I recommend accepting it for publication in the Chemistry journal.

Good luck.

Round 2

Reviewer 1 Report

(1)    For response of comment (1), I suggest the author double check the output files. Basis sets with effective core potentials do not indicate small basis sets. The number of basis functions for SDD is larger than 6-31g*. It is not acceptable to use large basis sets for optimization but small basis set for single point calculation. Besides, it is not time consuming to apply a large basis set for such a small system with only one metal atom.

(2)    For response of comment (3), "N atoms of toxa ligand are depeotonated; therefore, N atoms have an un-bonded lone pair that adds an additional coordination number to N atoms." is rather confusing. Does the author mean deprotonated? No lone pair was found in the revised manuscript and figure 6/7, although the author stated “To clarify our intention, we write lone pair on un-coordinated N atoms in Fig.6.” As strong C-N and N-H covalent bonds are present in toxa-(κS,N) and toxa-(κS,S) structures, the possibilities of lone pairs on N atoms is very small.
